# A cross-sectional survey on the effectiveness of public health campaigns for changing knowledge, attitudes, and practices in Kenyan informal settlements during the COVID-19 pandemic

**Steven Scholfield**[1], **Geraldine D. Kavembe**[2], **Rodney R. Duncan**[3], **Bernhards O. Ragama**[4], **Jared Mecha**[5], **Albert Orwa**[5], **Geoffrey Otomu**[6], **Erick Wanga**[7], **James Astleford**[7], **John Gutto**[1], **Isaac Kibwage**[8], **Julius Ogato**[9], **Arpana Verma**[10], **Keith Brennan**[1], **Jonathan Huck**[11], **Diana Mitlin**[12], **Mahesh Nirmalan**[1]*

1 Faculty of Biology, Medicine and Health University of Manchester, Manchester, United Kingdom, 2 Department of Life Sciences, South Eastern Kenya University, Kitui, Kenya, 3 Department of Monitoring and Evaluation, Central Kenya Conference of SDA, Nairobi, Kenya, 4 Centre for Research and Therapeutic Sciences, Strathmore University and Kenya Medical Research Institute (KEMRI), Nairobi, Kenya, 5 Department of Clinical Medicine and Therapeutics, University of Nairobi, Nairobi, Kenya, 6 Department of Medicine, Kisii Teaching and Referral Hospital, Kisii, Kenya, 7 Adventist Development and Relief Agency (ADRA), Nairobi, Kenya, 8 Egerton University, Nairobi, Kenya, 9 Division of Health Systems Strengthening, Ministry of Health, Nairobi, Kenya, 10 Division of Population Sciences, School of Health Sciences, University of Manchester, Manchester, United Kingdom, 11 MCGIS, Department of Geography, Faculty of Humanities, University of Manchester, Manchester, United Kingdom, 12 Global Development Institute, Faculty of Humanities, University of Manchester, Manchester, United Kingdom

* mahesh.nirmalan@manchester.ac.uk

## Abstract

We performed two cross-sectional surveys across three informal settlements in Kenya (within Kisii county, Nairobi, and Nakuru county) to study the effectiveness of public health interventions during the COVID-19 pandemic. A total of 720 participants were surveyed from 120 randomly selected geographical locations (240 participants/settlement/survey), and a coordinated health promotion campaign was delivered between the two surveys by trained staff. Information relating to knowledge, attitudes, and practices (KAP) were collected by trained field workers using a validated questionnaire. The main outcomes showed improvements in: (i) mask-wearing (% of participants 'Always' using their mask increased from 71 to 74%, and the percentage using their masks 'Sometimes' decreased from 15% to 6%; $p<0.001$); (ii) practices related to face mask usage (% of subjects covering the mouth and nose increased from 91 to 95%, and those covering only part of their face decreased from around 2.5% to <1%; $p<0.001$). Significant improvements were also seen in the attitudes and expectations relating to mask wearing, and in the understanding of government directives. Over 50% of subjects in the post-campaign survey reported that social distancing was not possible in their communities and fears associated with COVID-19 testing were resistant to change (unchanged at 10%). Access to COVID-19 testing facilities was limited, leaving a large proportion of people unable to test. As willingness to take a COVID-19 test

**Data Availability Statement:** All relevant data are within the manuscript and its Supporting Information files.

**Funding:** The study was funded by the Global Challenges Research Fund, QR allocations to the University of Manchester. The funders had no role in study design, data collection and analysis, decision to publish, or preparation of the manuscript.

**Competing interests:** The authors have declared that no competing interests exist.

did not change between surveys (69 vs 70%; $p = 0.57$), despite increased availability, we recommend that policy level interventions are needed, aimed at mitigating adverse consequences of a positive test. Improvements of KAPs in the more crowded urban environment (Nairobi) were less than at settlements in rural or semi-urban settings (Nakuru and Kisii). We conclude that coordinated public health campaigns are effective in facilitating the change of KAPs amongst people living amidst challenging socio-economic conditions in informal settlements.

## Introduction

### Public health interventions during a pandemic

Pandemics have had a significant role in shaping human history [1], and the recent COVID-19 pandemic was no exception. In addition to the novel aspects of SARS-CoV-2 pathophysiology [2], the pandemic also revealed and exacerbated the social and health inequalities that were inherent within many societies [3]. Amongst global communities, effective mass vaccination will likely play an important role in the decisive long-term control of future pandemics. However, vaccine development and deployment are time consuming, especially for low-and-middle-income countries (LMICs) who face additional constraints. In the interim, therefore, LMICs will need to rely on alternative, simple, and cost-effective public health measures to contain the spread of disease. When such public health measures involve limiting socio-cultural and economic activities (e.g., local travel restrictions and 'lockdowns'), and enforcing new behaviours (e.g., extended quarantine periods, face mask-wearing, and hand washing), the negative unintended consequences may also disproportionately affect the more vulnerable communities.

### Pandemics and living conditions

When all other variables are considered equal, the spread of respiratory pandemics will likely occur more rapidly in poor and over-crowded environments. In communicating important public health messages to people living under such conditions, bespoke and contextualised techniques may be needed. For example, in the context of accessing COVID-19 related public health messages, a recent study from Kenya highlighted the vulnerability of people with lower levels of education and socio-economic status, emphasising the need for context-sensitive strategies in communicating with these populations [4]. Given this background, the knowledge, attitudes, and practices (KAPs) of people living in such challenging and over-crowded 'informal settlements', and how they changed in response to the public health interventions during the COVID-19 pandemic, are relevant. As stated in the Alma-Ata declaration [5], 'health for all' can only be achieved through a bottom-up approach and espouse the principles of inclusivity in the implementation of potentially intrusive public health interventions in disease prevention.

### Informal settlements in Africa

'Informal settlements' and 'slum areas' have become the reflection of urban inequalities, poverty, and lack of access to adequate housing [6]. 'Informal' or 'spontaneous' settlements are human settlements whereby persons assert rights over land which is not registered in their names (i.e., land over which they do not have formal ownership or tenancy) [6]. The term

'slum', on the other hand, is a "general expression to describe a wide range of low-income settlements and/or poor human living conditions" [6]. Criteria used to define 'slum areas' include physical, spatial, social, and behavioural conditions, and the term itself carries negative connotations [6]. In reality, informal settlements and slum-like conditions often co-exist [7–10] and the two terms are often erroneously used interchangeably [11, 12]. In 2020, it was estimated that 51% of the sub-Saharan population to be living under such circumstances [13]. It is known that people living in informal settlements face a 'double burden' of health-related issues—a high risk of communicable and non-communicable disease [14]. The overcrowding and close physical proximity between household members, in many instances involving multiple generations, will likely facilitate the rapid transmission of droplet-borne respiratory diseases, as shown by a stochastic network model which found that peak and cumulative caseloads during a pandemic are higher for 'slum neighbourhoods than non-slum neighbourhoods' [15]. Moreover, residents in informal settlements can spend as much as ten percent of their daily income on water, often supplied by 'water barons' through informal vendors/kiosks [16, 17]. Such conditions make sanitation and hygiene measures extremely hard to practice in these neighbourhoods [18]. Similarly, basic sanitation and sewage disposal are poor [19], and the rates of malnutrition [20], chronic respiratory diseases (i.e., asthma and chronic bronchitis) [21], and HIV [22] are higher in these settings. Together, these factors can potentially render people living in informal settlements more susceptible to agents with pandemic potential. Furthermore, many residents often travel within and outside their neighbourhoods for work, further potentiating the rapid spread of disease within and beyond settlement lines [23].

Public health interventions usually work through positively influencing the KAPs of the populations concerned. Efficacy of KAPs are also partly related to the information put out by relevant authorities, the trust in those authorities [24], and the ease with which people can access information [25]. However, though there have been previous studies into KAPs in vulnerable populations in sub-Saharan Africa [23, 25–31], few attempts have been made to ascertain if these characteristics are positively modifiable through interventions. A rare example of such a study in Bangladesh, however, has demonstrated that one third of the population living in informal settlements maintained poor health practices with respect to COVID-19 transmission, and one quarter had poor knowledge of disease control [32]. Nevertheless, though similar challenges are known to exist within many informal settlements in sub-Saharan Africa, the magnitude of the problem is currently under-explored and poorly defined. The current study, therefore, summarises the findings from a public health campaign and two consecutive surveys at three informal settlements in Kenya. The study was performed to ascertain if a coordinated public health campaign, delivered by trusted local public providers, could have a positive impact on the KAPs known to be important in disease prevention.

## Materials and methods

### Study design

The KAPs around COVID-19 were studied in two cross sectional community surveys at three large informal settlements in Kenya in 2020. Survey participants were identified at 120 random geographical coordinates per settlement, selected using a spatial sampling approach, as previously described [33]. At each sample location, a minimum of two participants identified outdoors were surveyed using a pragmatic, stratified purposive sampling strategy, which permitted us to ensure adequate representation of age and gender. Following a baseline survey (July 2020) using a bespoke questionnaire, an intense and coordinated public health campaign was carried out over a 4-month period between August and November 2020. The health

campaign was delivered by a well-established international relief organisation (Adventist Development & Relief Agency, ADRA Kenya) with extensive experience working in low-income communities in Africa. The survey was repeated (after the intervention period) in December 2020 with a new set of random sample locations in order to determine the efficacy of the intervention.

## Survey locations and population estimation

The informal settlements selected for the study were located in Kisii county (Daraja & Nubian), Nairobi (Kibera), and Nakuru county (Rhoda). These were included due to a combination of their varied characteristics in terms of their size, setting (urban or rural) and population density, as well as ensuring that ADRA had sufficient community links and personnel for data collection. Due to the absence of detailed population surveys for these informal settlements and substantial variation in published estimates (estimates for Kibera range from 170,000 to over 1.5 million [34–36]), we adopted a standardised approach to population estimation based on the High Resolution Settlement Layer (HRSL) [37, 38] population dataset. This dataset comprises a surface of population density estimates at 1 arc-second (approximately 30m; resulting in approximately one value per 900m2) resolution, produced using a semi-automated process based on a combination of satellite imagery and census data. An estimated mean population density was calculated for each settlement, the bounds of which are manually digitised using high resolution satellite imagery, avoiding areas of non-residential land. The estimated mean population density is then multiplied by the residential area, resulting in an estimated population value.

Kibera is one of the most densely populated urban settlements in Kenya, with an estimated population of approximately 377,921 people in an area of approximately 2.5 km2 (133 people per 900m2). The Rhoda settlement in Nakuru is semi-urban and contains an estimated 279,446 people in an area of approximately 5.4 km2 (46.45 people per 900m2). The contiguous Daraja & Nubian settlements in Kisii are rural and contain an estimated 176,292 people in an area of approximately 21.5 km2 (7.4 people per 900m2). These population estimates are summarised in Fig 1.

## Development of survey questionnaire

A study questionnaire was developed through an iterative process with close involvement of ADRA, local collaborators, county public health workers and community volunteers at the three settlements. Focus group meetings held by ADRA involving local citizen volunteers, community leaders, local field health workers, health officials, and local authorities were used in framing the survey questions, ensuring the intended meanings of the questions were culturally appropriate and well understood by all stakeholders. Once agreed among various stakeholders, the questionnaire was further validated during focus group discussions at each settlement. The final questionnaire used in both surveys is provided in S1 File. The survey tools were completed digitally using the Kobo Toolbox software (https://www.kobotoolbox. org), either directly by the surveyed subjects, or with assistance from trained field assistants fluent in the local language.

## Sampling strategy

Sampling targets were determined using a quota sampling approach, in which a set number of stratified samples was collected in each settlement. Our power calculations [39] specified at least 114 samples per site were required, but to account for the high uncertainties associated with both the population value (as described above) and likely levels of prevalence, we elected

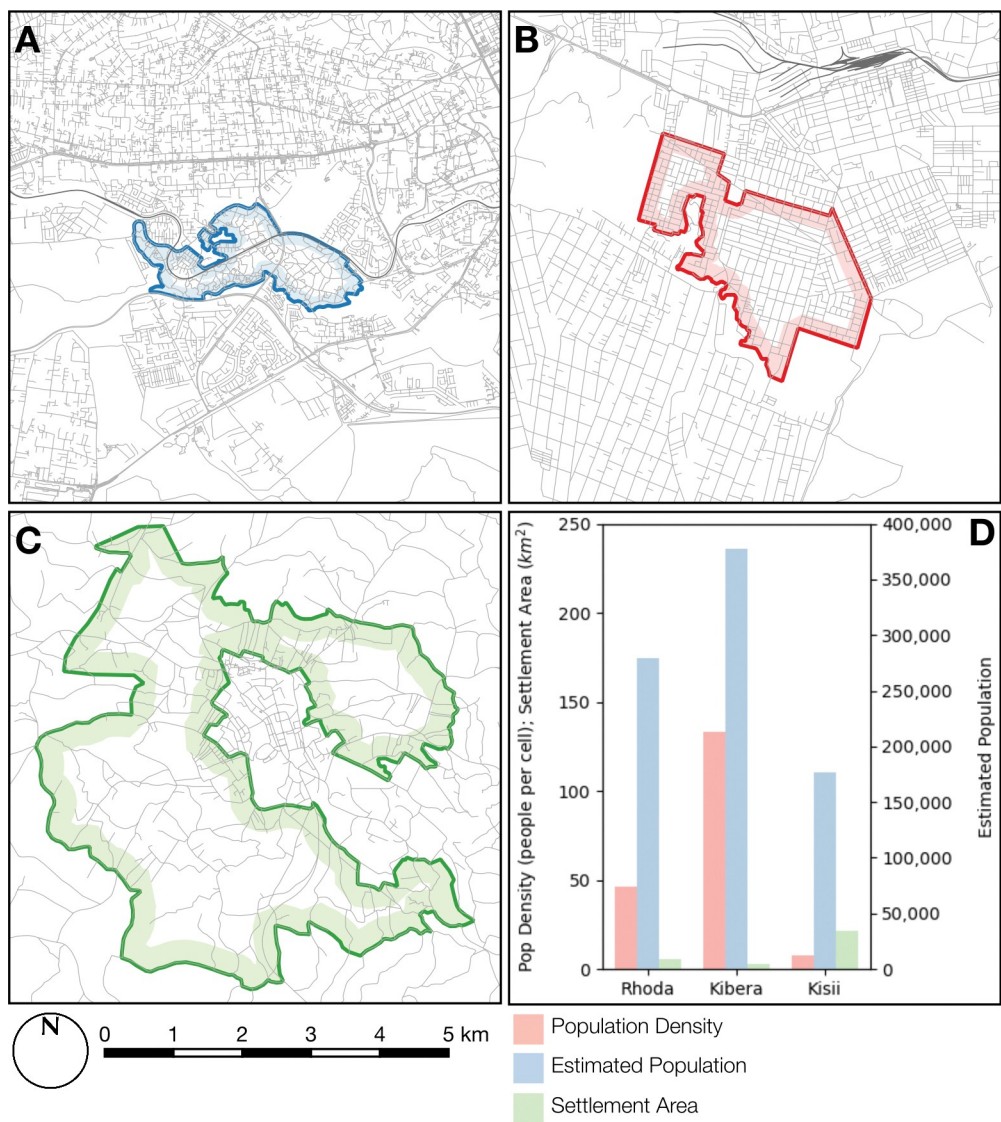

**Fig 1.** Informal settlements areas included in the study: A. Kibera (urban); B. Rhoda (semi-urban) and C. Daraja & Nubian (rural). D. Estimated population density and estimated total population for the respective settlements. Road data © 2023 OpenStreetMap Contributors.

to double this to 240 per location (this approach was also used by [33] when undertaking prevalence studies in challenging environments with limited data), and used the same sample size for each of the settlements. These quota sampling targets were stratified for each settlement using data from the Kenyan national census, which allowed us to ensure that a representative sample of the population was achieved, based on local proportions of sex and age reported in the national census. Sampling locations in each settlement were selected using a simple pseudo-random point generator in QGIS version 3. A total of 120 unique pseudo-random points were generated within the residential bounding polygon (described above) for each settlement, at which two individuals are to be sampled, resulting in 240 samples in total. At each sampling location, the first two or three volunteers who were encountered chronologically and willing to take part in the survey were recruited to the study.

## Health promotion and the conduct of field work

All field workers underwent training in questionnaire administration, data collection, and data handling, of which was facilitated by ADRA Field Educators and Regional Health Officials from local health departments. ADRA has an extensive experience in providing training to field workers and the infrastructure available in-house was utilised for the delivery of these training sessions. A minimum of two training workshops was provided to all field workers at each settlement. The field workers were selected locally, ensuring acceptability and ease of data collection due to the use of the local language in the administration of questionnaires. All data was collected using anonymised codes and the study team did not collect or have access to identifiable personal data at any stage of the study. The health promotion activities (S2 File) included media campaigns through local radio stations, home visits, health talks at places of worship, barazas (public awareness raising and sensitisation forums), meeting places, bus parks, distribution of health promotion leaflets at common community venues, and the provision of hand sanitisers and approximately 60,000 locally sourced reusable face masks. The distribution of sanitiser and masks were carried out across the community as part of the campaign and was independent of the two surveys to ensure that the survey responses were not influenced by the incentives of free masks, sanitisers, or information leaflets. Community health volunteers (CHVs), community leaders, influencers, local administration officials (i.e., chiefs), and village elders were selected from each settlement and facilitated meetings to enable higher co-operation from the respective communities.

The CHVs who were part of the study teams were also part of the community health service delivery mechanism within the counties, and they took part in the delivery of these interventions. The CHVs worked closely with ADRA staff and county public health officers to ensure the validity of messaging and to limit distortion and misrepresentation of facts. ADRA recruited and trained 30 dedicated CHVs (10 per settlement) to deliver the promotion campaign. The recruited CHVs were briefed on the project goal and scope, adherence to standard operating procedures, and personal protection. Bespoke brochures, leaflets, and flyers (developed in partnership with the local health officials) were distributed free of charge. Wherever possible, the local radio stations, other media (such as local music shops), and social media were co-opted into the campaign to broaden access. The public health campaigns conducted by the regular government public health officials–utilising similar methodologies and other social media platforms continued unimpeded by the focussed campaign carried out as part of the present study.

## Monitoring of field study

An independent quality assurance (QA) team regularly undertook field checks on the conduct of the campaign and adherence to basic safety guidelines. Feedback from the QA team was passed to the formal local study governance (LSG) group, which met weekly to review progress and mitigate any local challenges. Feedback from the QA team and the observations of the LSG group were used iteratively to ensure that the health promotion campaign reached out across the entire settlement and was not localised to any particular area based on convenience or ease of access.

## Statistical analysis

All data were tabulated by response, and exported to IBM SPSS Statistics, Version: 28.0.1.0 (142) for statistical analysis. The main outcomes were the changes in KAP measures before and after the public health campaign. Categorical data were assessed using chi-square tests, and continuous data were analysed with Student's t-test. Ordinal logistic regression was used to assess for the interacting effect of participant characteristics and settlement type on the

changes on KAP measures before and after the public health campaign. The significance level was defined at .05 for all tests.

## Results

Although our questionnaire contained 51 items, for the purposes of brevity and clarity, only the pertinent quantitative data is presented for analysis in this manuscript. For each survey, we targeted 720 individual participants from 360 randomly selected geographical coordinates across the three settlements. In the two surveys, we received 1,413 individual responses, however 39 of these were excluded from data analysis, owing to the following reasons: 28 had unfeasible answers (e.g., working >7 days per week or >24 hours per day), 6 had contradictory answers, 3 did not consent to inclusion in statistical analysis, and 2 had inappropriate answers for the posed question (i.e., suggesting that they had misread or misunderstood the question). The distribution of excluded responses across the three settlements were as follows: Survey 1: Kisii 5, Kibera 13 and Nakuru 9, and Survey 2: Kisii 1, Kibera 3 and Nakuru 5. Therefore, a total of 1374 individual responses were analysed, with 691 and 683 people in the first and second surveys, respectively. Due to the stratification in the sampling method, there were no differences in the distribution of people or gender across settlements. Any data that fell outside of the outlier threshold for continuous data pertaining to mask price (>3 standard deviations about the mean) were not included for calculating mask prices. The Survey 1 group was slightly older in years, with a mean age ± standard deviation (SD) of 40.0 ± 14.9 vs. 37.7 ± 13.4 ($p$ = 0.003). There were also small differences in educational backgrounds as collectively more people had attended secondary school or college in survey 2 (55% vs. 65%). These data are summarised in Table 1.

### Mask use and procurement behaviours

The proportion of participants who reported that they wore a mask rose in Survey 2 from 97.7% to 99.1%. Differences were also seen in the frequency of wearing masks when going out,

**Table 1. Data showing the collated participant demographics for all three settlements.**

| | Total settlement data | | |
| --- | --- | --- | --- |
| | Survey 1% (counts) | Survey 2% (counts) | *p*-value |
| Settlement population | | | |
| Percentage | 50.3 (691) | 49.7 (683) | — |
| Age | | | |
| Years (SD; Range) | 40.0 (14.9; 18–90) | 37.7 (13.4; 18–95) | 0.003* |
| Gender | | | |
| Female | 50.9 (352) | 50.1 (342) | 0.577 |
| Male | 49.1 (339) | 49.8 (340) | |
| Other | 0.0 (0) | 0.1 (1) | |
| Highest educational level | | | |
| No education | 4.9 (34) | 4.5 (31) | 0.006* |
| Primary school | 31.5 (218) | 24.0 (164) | |
| Secondary school | 40.4 (279) | 44.1 (301) | |
| College | 15.1 (104) | 20.9 (143) | |
| University | 7.2 (50) | 5.6 (38) | |
| Postgrad. degree | 0.9 (6) | 0.9 (6) | |

Values are shown as percentages (counts) unless otherwise stated. SD: standard deviation.

* Significant at 5%

**Table 2. Data showing the behaviours in mask use, procurement, maintenance, and training of *all* settlement data, collated.**

| | Total settlement data | | |
| --- | --- | --- | --- |
| | Survey 1% (counts) | Survey 2% (counts) | *p*-value |
| Do you wear a mask? | | | |
| Yes | 97.7 (675) | 99.1 (677) | 0.034* |
| No | 2.3 (16) | 0.9 (6) | |
| Frequency of mask wearing when out | | | |
| Always | 70.7 (477) | 73.9 (500) | <0.001* |
| Most of the time | 14.2 (96) | 19.9 (135) | |
| Sometimes | 15.1 (102) | 6.2 (42) | |
| Part of face covered when mask wearing | | | |
| Mouth and nose | 91.0 (629) | 95.0 (649) | <0.001* |
| Mouth only | 2.6 (18) | 1.0 (7) | |
| Nose only | 2.6 (18) | 0.1 (1) | |
| Varies | 3.8 (26) | 3.8 (26) | |
| Frequency of mask re-positioning | | | |
| Never | 6.1 (41) | 4.1 (28) | <0.001* |
| Infrequently | 15.9 (107) | 41.8 (283) | |
| Frequently | 46.7 (315) | 30.0 (203) | |
| Very Frequently | 31.4 (212) | 24.1 (163) | |
| Source of mask | | | |
| Government / NGO | 11.2 (80) | 37.0 (302) | <0.001* |
| Shop | 84.0 (600) | 57.4 (468) | |
| Internet | 1.4 (10) | 2.2 (18) | |
| Self/family-made | 3.4 (24) | 3.4 (28) | |
| Mask / mask material price (KES)a | | | |
| Amount in KES (SD) | 52.90 (32.45) | 33.21 (23.88) | <0.001* |

Values are shown as percentages % (counts) unless otherwise stated. KES: Kenyan Shilling; SD: standard deviation.

a Sample size of respondents for each settlement: Kisii: 330; Nairobi: 372; and Nakuru: 398.

* Significant at 5%

with people reporting '*Always*' increasing from 71% to 74%, '*Most of the time*' increasing from 14.2% to 19.9%, and '*Sometimes*' decreasing from 15.1% to 6.2%. The percentage of patients applying their masks to cover their mouth and nose rose from 91% to 95%. Similar positive behavioural changes were observed in the frequency at which people repositioned their masks while in use (Table 2). Sources from where people obtained their masks also changed in Survey 2, with increased reliance on government sources (11% to 37%) than private shops (84% to 57%). These shifts were quantitatively substantial and statistically significant (see Table 2). Along with the shift in the source of face masks, a reduction in the cost/day was also seen (52 KES/day to 33 KES/day) These data are summarised in Table 2.

## Behaviours around mask maintenance and training

Fewer reusable (79% to 54%), and more disposable (12% to 25%), face masks were reported in survey 2. Of the people who persisted with reusable masks, the proportion of subjects washing their masks daily increased by approximately 10% (57.8% to 68.3%). These positive changes in subject behaviour were associated with almost a six-fold increase in the number of subjects reporting they had received adequate information or training on the correct use and

**Table 3. Data showing the behaviours around mask use, procurement, maintenance, and training of *all* settlement data, collated.**

| | Total settlement data | | |
| --- | --- | --- | --- |
| | Survey 1% (counts) | Survey 2% (counts) | *p*-value |
| Are your masks reusable or not | | | |
| Reusable | 79.0 (562) | 54.4 (394) | <0.001* |
| Disposable | 12.0 (85) | 25.3 (183) | |
| Sometimes both | 9.0 (64) | 20.3 (147) | |
| Maintenance of reusable masks | | | |
| Washing | 98.6 (554) | 99.0 (390) | 0.768 |
| Dry cleaning | 0.4 (2) | 0.3 (1) | |
| Antiseptic dipping | 0.4 (2) | 0.5 (2) | |
| None of the above | 0.7 (4) | 0.3 (1) | |
| Washing frequency of reusable masks | | | |
| Daily | 57.8 (325) | 68.3 (269) | 0.013* |
| 2–3 days | 30.6 (172) | 23.6 (93) | |
| Weekly | 8.4 (47) | 6.9 (27) | |
| Fortnightly | 0.7 (4) | 0.8 (3) | |
| Monthly | 2.1 (12) | 0.5 (2) | |
| Never | 0.4 (2) | 0.0 (0) | |
| Frequency of replacing reusable masks | | | |
| Daily | 27.4 (154) | 21.8 (86) | 0.008* |
| 2–3 days | 23.5 (132) | 28.9 (114) | |
| Weekly | 18.0 (101) | 19.5 (77) | |
| Fortnightly | 4.3 (24) | 5.1 (20) | |
| Monthly | 14.2 (80) | 17.8 (70) | |
| Never | 12.6 (71) | 6.9 (27) | |
| Have you had any information or training provided on the correct use of masks | | | |
| Adequate information | 11.3 (78) | 69.4 (474) | <0.001* |
| Some information | 41.1 (284) | 26.2 (179) | |
| None | 46.9 (324) | 4.1 (28) | |
| N/A | 0.7 (5) | 0.3 (2) | |

Values are shown as percentages % (counts) unless otherwise stated.

*Significant at 5%

maintenance of face masks in the intervening period (11% to 69%). These data are summarised in Table 3.

## Attitudes to mask wearing and perceived effectiveness

Attitudes to mask wearing also changed significantly between the two surveys. Comparatively, more people in Survey 2 expected those with a cough to wear a mask (83% to 89%), and a greater proportion believed that masks were useful in controlling the spread of COVID-19. The latter is evidenced by the shift in responses to the question '*Do you believe that wearing a mask is useful to prevent the spread of COVID-19 pandemic?'*. In response to this question, the number of people who answered '*Definitely'* increased by more than 10% (80.3% to 90.9%), and the percentage of people who answered either '*Possibly'* or '*Do not know*' collectively reduced by more than 10%. Over, 50% of subjects in Survey 1 reported concerns that the use of face masks may actually be harmful at a time of a respiratory pandemic. The corresponding figure reduced to c.38% in Survey 2. These data are summarised in Table 4.

**Table 4. Data showing the attitudes to mask wearing and perceived effectiveness of *all* settlement data, collated.**

| | Total settlement data | | |
|---|---|---|---|
| | Survey 1% (counts) | Survey 2% (counts) | *p*-value |
| Do you expect a person with chronic cough to use a face mask? | | | |
| Yes | 82.5 (570) | 88.6 (605) | <0.001* |
| No | 14.6 (101) | 6.4 (44) | |
| Uncertain | 2.9 (20) | 5.0 (34) | |
| Do you believe that wearing a mask is useful to prevent the spread of COVID-19? | | | |
| Definitely | 80.3 (555) | 90.9 (621) | <0.001* |
| Possibly | 17.4 (120) | 7.8 (53) | |
| Do not know | 2.3 (16) | 1.3 (9) | |
| Do you feel that use of mask can be harmful at a time of a respiratory pandemic? | | | |
| Yes | 53.5 (370) | 37.6 (257) | <0.001* |
| No | 38.8 (268) | 54.5 (372) | |
| Uncertain | 7.7 (53) | 7.9 (54) | |

All values are shown as percentages % (counts).

*Significant at 5%

## Behaviours and attitudes to COVID-19 testing

Despite the fact that COVID-19 tests were not freely available and there were significant cost implications to having these tests done, there was almost a doubling of the percentage of subjects who had a COVID test between the two surveys (approx. 10% to 20%). However, in response to the question '*Would you be willing to have a COVID-19 test if it were available*' approximately a third of the subjects answered '*No*' at both surveys (30.8% vs. 29.4%). This apparent unwillingness to take the COVID-19 test was associated with persistent concerns over well recognised barriers such as '*Fear of quarantine*', '*Fear of a positive test*', '*Stigma*' etc. remaining unchanged between the two surveys. These data are summarised in Table 5.

## Behaviour and attitude towards social distancing and other government directives

There were significant changes in people's responses as to what the ideal distance should be to achieve effective social distancing. The number of people who said 1.5 metres (recommended by the government of Kenya) [40] increased from approximately 41% to 58%, whilst the percentage of people who said this distance to be one metre decreased from approximately 42% to 28%. However, more than 50% of responders in the second survey stated that social distancing was not possible in their daily lives. Awareness of government-imposed travel restrictions was high from the beginning (>95% of responders at both surveys) and the belief in the value of such restrictions increased from approximately 69% to 74% between the two surveys. During the same period, the mistrust in such policies decreased from 15.5% to approximately 4%. These data are summarised in Table 6.

## Effect of participant characteristics and informal settlement on mask wearing practices

We used ordinal logistic regression to assess the effect of participant characteristics and informal settlement on mask wearing practices ranging from responses '*Always when I go out*' to '*Never*'. Participants residing in Rhoda (Nakuru), had 6.21 higher odds of increased mask

**Table 5. Data showing the behaviours and attitudes to COVID-19 testing of *all* settlement data, collated.**

| | Total settlement data | | |
| --- | --- | --- | --- |
| | Survey 1% (counts) | Survey 2% (counts) | *p*-value |
| Have you been tested for COVID-19? | | | |
| Yes | 9.8 (68) | 20.1 (137) | <0.001* |
| No | 90.2 (623) | 79.9 (546) | |
| Reasons test not taken | | | |
| Fear of quarantine | 8.3 (52) | 6.2 (34) | 0.516 |
| Fear of positive test | 10.4 (65) | 9.2 (50) | |
| Not sick | 15.2 (95) | 15.4 (84) | |
| Stigma | 4.5 (28) | 3.8 (21) | |
| Not available | 61.5 (383) | 65.4 (357) | |
| Would you be willing to have a COVID-19 test if it were available? | | | |
| Yes | 69.2 (478) | 70.6 (482) | 0.573 |
| No | 30.8 (213) | 29.4 (201) | |

All values are shown as percentages % (counts).

*Significant at 5%

wearing in Survey 2 compared to participants residing in Kisii (Daraja & Nubian) and Kibera (Nairobi). Moreover, for each step in educational attainment, participants with a higher educational background had incrementally greater odds of wearing a mask. These data are summarised in Table 7.

**Table 6. Data showing the attitudes to social distancing, and government directives of *all* settlement data, collated.**

| | Total settlement data | | |
| --- | --- | --- | --- |
| | Survey 1% (counts) | Survey 2% (counts) | *p*-value |
| How many metres for social distancing | | | |
| 1 metre | 42.4 (293) | 28.0 (191) | <0.001* |
| 1.5 metres | 41.0 (283) | 57.5 (393) | |
| 2 metres | 14.2 (98) | 13.9 (95) | |
| Do not know | 2.5 (17) | 0.6 (4) | |
| Can people socially distance in your community | | | |
| Yes | 31.7 (219) | 48.6 (332) | <0.001* |
| No | 68.3 (472) | 51.4 (351) | |
| Awareness of government COVID-19 related restrictions of travel | | | |
| Yes | 97.1 (671) | 97.8 (668) | 0.171 |
| No | 2.0 (14) | 2.0 (14) | |
| Don't know | 0.9 (6) | 0.1 (1) | |
| Perception of government COVID-19 directives | | | |
| Obedience | 16.2 (122) | 21.4 (169) | <0.001* |
| Believe them | 66.8 (504) | 73.7 (581) | |
| Mistrust | 15.5 (117) | 4.3 (34) | |
| Useless | 1.5 (11) | 0.5 (4) | |

All values are shown as percentages % (counts)

*Significant at 5%

**Table 7. Data showing results of logistic regression analysis of participant characteristics and settlement type on changes in mask wearing.**

| | Coefficient (SE) | Adjusted OR (CI) | *p*-value |
|---|---|---|---|
| Survey Time | | | |
| Baseline | Reference | | |
| After campaign | 0.892 (0.329) | 2.44 [1.28–4.65] | 0.007* |
| Settlement area | | | |
| Daraja/Nubian, Kisii | Reference | | |
| Kibera, Nairobi | 0.255 (0.282) | 1.29 [0.74–2.24] | 0.367 |
| Rhoda, Nakuru | 1.826 (0.522) | 6.21 [2.23–17.26] | <0.001* |
| Gender and Age | | | |
| Female | Reference | | |
| Male | -0.042 (0.126) | 0.96 [0.75–1.23] | 0.740 |
| Age | -0.001 (0.005) | 1.00 [0.99–1.01] | 0.770 |
| Education level | | | |
| Never attended education | Reference | | |
| Primary school | 0.631 (0.293) | 1.88 [1.06–3.34] | 0.032* |
| Secondary school | 0.685 (0.296) | 1.98 [1.11–3.55] | 0.021* |
| College | 0.724 (0.318) | 2.07 [1.11–3.85] | 0.023* |
| University degree | 0.866 (0.378) | 2.43 [1.16–5.09] | 0.019* |
| Postgraduate degree | 2.142 (1.087) | 8.52 [1.01–71.73] | 0.049* |

SE: standard error; OR: odds ratio.

*Significant at 5%

### Data related to participant characteristics and responses at individual settlements

The above analyses repeated with data disaggregated by settlement are given in S1-S5 Tables in S3 File. The patterns observed within these individual settlements were broadly similar as reported above for the entire cohort, even though there is some suggestion that the magnitude of changes seen in the urban Kibera settlement was less than the corresponding changes at the semi-urban Rhoda settlement in Nakuru, or the more rural Daraja & Nubian settlements in Kisii (for some of the key questions in the survey). For example, responses to the question '*How frequently subjects wore the mask when leaving home*?' saw the percentage of people who responded '*Always*' increase by 15% at Kisii and by 6% at Nakuru but fall by almost 10% at the Kibera settlement (S2a Table in S3 File). Similarly, the change in responses to '*Do you believe that wearing a mask is useful to prevent the spread of COVID-19*?' was more muted at Kibera than at either Rhoda or Daraja & Nubian (Table 4).

### Discussion

In these cross-sectional surveys conducted at two distinct time-points, we assessed the changes in KAPs before and after a public health campaign across three informal settlements in Kenya during the COVID-19 pandemic. Apart from one previous study carried out during a similar period in rural Bangladesh [41], there is a paucity of data to understand the effectiveness of public health interventions within low-income informal settlements. Even though many authors have explored KAPs in vulnerable populations living in Africa at a single time point [23, 25–31], few attempts have been made to ascertain if these characteristics are positively modifiable. Our study adds to the current knowledge base by providing data across multiple domains relating to changes in KAPs when an effective public health intervention is

implemented in informal settlements. The socio-cultural, economic and logistical challenges faced by residents of informal settlements in Africa (and other LMICs) are unique. An objective understanding of how people living under such challenging conditions respond to health messages during a public health emergency is important to public health officials and policy makers, not only for now, but also in planning their responses to similar challenges in the future.

The main outcomes showed improvements in: (i) mask wearing behaviours, (ii) mask maintenance, (iii) attitudes and expectations relating to mask wearing, and (iv) attitudes, knowledge, and wider understanding of government directives. Social distancing and the fears associated with volunteering for COVID-19 testing were more resistant to change, suggesting the need for specific policy interventions to address these concerns. Intrinsic informal settlement characteristics (urban, peri-urban, rural, population density, resident characteristics, previous experience with government and NGOs etc.) could have a moderating effect on KAPs towards public health measures, hence the need to contextualise the delivery of the measures.

The study population was relatively young at both time points, (mean age 40 vs. 37.7; $p = 0.003$; Table 1), which reflects the population of Kenya (as is expected with a stratified study). Though the change in age distribution between the two surveys is small (2.3 years), when combined with the associated reduction in the standard deviation of the ages in the second survey, it is possible that this could be indicative of the more elderly population not venturing out as frequently by the time of the second survey, resulting in a younger and more homogeneous age distribution in the second survey sample.

The approach adopted for our health campaign draws widely from accepted behaviour change principles that can help control transmission of airborne infections in the population [42]. Local government health workers and CHV representation at training sessions ensured that our messaging was consistent with the broad principles adopted by the Kenyan government and the county health authorities. During this period, the Kenyan government implemented interventions including daily curfews, social distancing (minimum of 1.5m), mask-wearing, hand sanitation, and COVID-19 testing when possible through a variety of means [40, 43, 44]. At times, strict punitive measures were employed by police to ensure compliance in response to the rapidly changing transmission rates, with spikes of COVID 19 cases being recorded in early August and in mid-November [45, 46]. The number of COVID-19 tests that were performed in the country also saw a steep rise during this period [47, 48]. Even though attempts were made to ease some measures nationwide in August, movement restrictions were re-imposed in November 2020, coinciding with a rapid rise in case numbers [44]. The guidance issued jointly by the Africa CDC and African Union [49, 50] was the basis of policy making in Kenya, which though comprehensive, did not make any provisions for daily wage earners (mostly casual labourers) living in informal settlements. No part of society in Kenya therefore could have been totally untouched by these intercessions, and this would account for the high awareness of government restrictions in the pre-intervention cohort (97.1%), the rise in disposable mask use and the fall in mask costs (with increased mask availability) that we observed in our study.

In Kenya, health services are delivered by devolved sub-national county governments, and despite central directives from the Ministry of Health in Nairobi, the implementation of these directives was in the hands of the county governments, with varying levels of funding allocated for public health. In our experience, engagement with health policy is historically low within informal settlements. That said, many civil society organisations were facilitating COVID-19 related measures in informal settlements during the outbreak [51], and Nairobi-based CHVs were to receiving monthly stipends [52] and such interventions may have facilitated greater

engagement. Our intervention (a public health campaign consistent with the WHO and Africa CDC recommendations) [49, 50] is most likely to have had a similar impact of ensuring that government policy was implemented effectively on the ground. It also may have had an additional 'dose effect' through relieving some of the economic burden that individuals/families may have faced, thereby reinforcing the changes that were occurring through government interventions. As a control group is not included, it is not possible to estimate the absolute impact of our intervention, as the observed changes may simply reflect the natural changes that were happening in society as people became more aware of the pandemic and its consequences. Systematic inclusion of a control group in a study involving some of the most marginalised communities would have posed serious ethical challenges, and hence was not considered in the study design.

In the previous study from Bangladesh, where the authors demonstrated a large (approximately 30%) increase in mask wearing following the health promotion campaign, the baseline mask usage was significantly lower (around 13%) [41]. In comparison, our baseline mask wearing was very high (> 95% across all three settlements). Despite this high baseline, small but significant improvements were still observed. This finding is important as even small gaps in adherence can facilitate the spread of diseases such as COVID-19 [53]. This implies that ongoing public health campaigns can have a beneficial effect, even when high levels of coverage have been achieved. The increase in the percentage of subjects who used masks as recommended, increase in frequency of mask wearing, reduction in physical handling of masks (Table 2), and the improvement in knowledge of mask-related benefits (Table 4) adds further support that the public health campaign was likely contributing to positive behaviour change. Notably, appropriate mask use is important for three reasons. First, it is implicit that only consistent and appropriate mask use will provide the greatest benefit to the individual and the population. Second, it is also self-evident that greater mask efficacy will come from covering the mouth and nose. Third, frequent mask handling and repositioning would likely reduce the benefits associated with mask use [54]. Similarly, repeated handling may increase the risk of contamination and secondary infection through touching. Such considerations, therefore, may explain why the benefits of face masks only become apparent in conjunction with hand hygiene [55, 56]. In our study, we also successfully established a local supply chain which was able to supply 60,000 reusable masks that were distributed free of charge. The distribution of masks was done by an independent team across the settlements and hence did not influence the conduct of the survey or its findings. Opportunities to mobilise such local supply chains should be seized by the relevant authorities as they may be the only source of supply and also provide livelihood assistance for local residents.

Whilst our study showed a small increase in the number of people who wore a mask (97.7% to 99.1%, $p$ = 0.034), the more substantial findings appear to be around changes in mask type. A relatively large change was observed in the type of masks used at the two time points, with almost a 25% reduction in the use of reusable masks and a large increase (approximately 15%) in the use of disposable masks (Table 3). These observations would suggest that, with the greater availability and reduced costs of disposable masks, people preferred the convenience of these over reusable masks, as the latter requires regular washing. The fact that this preference was seen despite the distribution of reusable cloth masks (free of cost) across all three settlements is highly relevant. The preference of disposable masks will clearly raise environmental issues in relation to their safe disposal. As highlighted previously, the environmental impact of reusable cloth masks are significantly lower than disposable masks [57, 58], particularly in low resource environments that lack clear policies and facilities for solid waste disposal [59]. Of those who continued to wear a reusable cloth mask, we found a 10% increase in the proportion of subjects who washed their masks every day. Interestingly, this increase was associated with

an almost 6-fold rise in the percentage of participants who had received adequate training in the use of face masks. This would suggest that there was greater awareness within the community on the importance of mask hygiene; a critical consideration when using reusable masks (Table 3) to reduce the risk of self-contamination [60].

Starting from a low baseline of approximately 10%, we observed a doubling of subjects who had taken a COVID test (PCR and/or lateral flow-based tests) between the two surveys (Table 5). The majority (>60%) attributed the unavailability of COVID tests as the reason for not doing the test, and this reflects the limited testing capability of many African states during the early days of the pandemic, owing to inadequate number of testing kits [61]. However, the near absence in any shift in response to the question 'Would you be willing to have a COVID-19 test if it were available?' (69.2% vs. 70.6%, $p = 0.573$) would suggest that there were still persistent concerns within the community, and the reasons for the relatively poor uptake of the COVID-19 test was unlikely to be attributable to availability alone. Other reasons for not getting a test, such as fear of consequences such as mandatory quarantine (approximately 15–20%), not feeling sick (15%) and social stigma (approximately 5%), were also mentioned by a sizeable minority (Table 5). The fact that the percentage of subjects who cited these 'other' reasons did not change suggests that the public health campaign did not sufficiently alleviate the fears and concerns of a small but sizeable minority of people. It also could imply that the socio-economic consequences of a positive test were potentially prohibitive for people to volunteer for testing. The need to increase access to COVID-19 tests in Africa has been highlighted during the pandemic [61, 62], and our surveys suggest that increasing access alone, without suitable policy frameworks to address the consequences of a positive test, may not lead to the desired outcomes.

In terms of social distancing, there was a very high level of awareness on the government-imposed travel restrictions at both time points of the survey (>95%). Encouragingly, approximately 20% more subjects felt that social distancing was possible at the second survey. Despite these positive changes, almost 50% of people in the post-intervention survey still believed that social distancing was not possible in their environments, which is concerning. Adherence to government directives also increased by 6.9% and 5.2% respectively, with an 11% fall in mistrust. This is pivotal as greater trust on government policies and guidelines are essential in the success of public health programmes [63–65].

This study shows that participant educational attainment had a moderating effect on the odds of mask wearing (Table 7). Although not all previous studies have found an effect of educational status on adherence to protective measures in the context of pandemics or COVID-19 [66], our findings support the hypothesis that persons with higher educational attainment were more willing to comply with preventative measures in this context [67–69]. The definitive mechanism remains unclear, but these results may reflect a greater ability to interpret and understand a variety of public health messaging around the virus [66, 67, 70, 71]. Moreover, as our intervention, in part, relied on leaflets to disseminate information, those who could not read-and-write (i.e., those with a lower educational background) would have been comparatively less influenced by the promotion campaign than more literate counterparts.

Although the present study was not designed to assess distinctions between settlements, the post-hoc differences that are evident cannot be overlooked (S1-S5 Tables in S3 File). The general direction of change in KAPs was similar across all three settlements. In all the 21 presented questions, there were 16 and 14 significant favourable changes in Kisii and Nakuru, respectively ($p<0.05$), and only 8 favourable changes in Kibera ($p<0.05$). These differences are also apparent in the logistic regression data shown in Table 7. Together these findings would suggest that the health promotion campaigns were less effective in changing KAPs within Kibera —the most crowded and most cosmopolitan urban settlement. The reasons for this could

perhaps relate to differences in population density, the urban setting, quality of life, pre-existing levels of NGO activity (which has historically been much higher in Kibera than the other settlements) or levels of trust in the local public health authorities. These potential inter-settlement differences are important in planning future programs, and warrant further investigation.

## Limitations

Some potential limitations in our study need emphasis. First, due to lack of control groups, as explained above, and the cross-sectional nature of the surveys, we cannot be certain of a causal relationship between the changes that occurred and the public health interventions that were provided. Secondly, despite our integrated and iterative process of developing the study questionnaire, an oversight occurred in the responses for questions 12 and 13 (S1 File). This only became apparent during data analysis. Responses measured the daily number of contacts in intervals, although some of these intervals overlapped. Subsequent answers, therefore, could have fallen into two separate categories, meaning we could not accurately stratify responses to their matched category. Third, because the present study used a questionnaire, the accuracy of some data was dependent on participant candidness or recall. Some responses may have been over-or-understated toward the outcome perceived as most appropriate. For example, our study showed significant disparities in mask compliance to another study in Bangladesh [41], which used a direct-observation approach to measure mask compliance, eliminating any overestimation from participants. This is relevant, as self-reporting in Kenya has been shown to overestimate rates of mask compliance [72] and may explain why our mask compliance was so high (97.7% and 99.1%). Equally, some results may have been subject to recall bias, particularly for quantitative data and questions pertaining to *time* (i.e., time spent at home on a workday). Finally, our sampling strategy was dependent on the first two or three subjects who were approached consenting to take part at every random geographical coordinate. As this was a voluntary choice, it is possible that only those subjects who had time, knowledge, inclination and civic mindedness to understand the importance of such surveys within their communities were recruited. This may account for some of the positive behavioural changes reported in our study. The use of ADRA–a locally trusted NGO, and CHVs who were embedded within each community minimised this risk as most subjects who were approached consented and took part in the study.

## Conclusion

In summary, our findings imply that well conducted public health campaigns are effective in changing the KAPs amongst people living within informal settlements. However, their ability to change may be limited by several constraints that are unique to such settlements. Moreover, COVID-19 testing availability remained an issue throughout our study interval, leaving a large proportion of people unable to test, despite their motivations. As willingness to take a COVID-19 test did not change over time, we also suggest that policy makers should focus on broadening access, whilst combating the anxieties faced by the community that prevent acceptance of as preventive measure. Finally, we observed potential differences between settlements which require further investigations to understand if the contextual factors unique to individual settlements could affect the effectiveness of coordinated public health action during a public health emergency.

## Supporting information

**S1 File.**
(PDF)

**S2 File.**
(DOCX)

**S3 File.**
(DOCX)

## Author Contributions

**Conceptualization:** Bernhards O. Ragama, Jared Mecha, Isaac Kibwage, Arpana Verma, Diana Mitlin, Mahesh Nirmalan.

**Data curation:** Geraldine D. Kavembe, Jared Mecha, Albert Orwa, Erick Wanga, Jonathan Huck, Mahesh Nirmalan.

**Formal analysis:** Steven Scholfield, Geraldine D. Kavembe, Jared Mecha, Albert Orwa, Jonathan Huck, Mahesh Nirmalan.

**Funding acquisition:** Jared Mecha, Keith Brennan, Diana Mitlin, Mahesh Nirmalan.

**Investigation:** Geraldine D. Kavembe, Rodney R. Duncan, Bernhards O. Ragama, Jared Mecha, Erick Wanga, Mahesh Nirmalan.

**Methodology:** Bernhards O. Ragama, Jared Mecha, Erick Wanga, Arpana Verma, Jonathan Huck, Diana Mitlin, Mahesh Nirmalan.

**Project administration:** Geraldine D. Kavembe, Rodney R. Duncan, Bernhards O. Ragama, Jared Mecha, Erick Wanga, John Gutto, Isaac Kibwage, Mahesh Nirmalan.

**Resources:** Erick Wanga, James Astleford, John Gutto, Isaac Kibwage, Julius Ogato, Arpana Verma, Keith Brennan.

**Software:** Steven Scholfield, Jonathan Huck.

**Supervision:** Rodney R. Duncan, Bernhards O. Ragama, Jared Mecha, Geoffrey Otomu, Erick Wanga, James Astleford, John Gutto, Isaac Kibwage, Julius Ogato, Mahesh Nirmalan.

**Validation:** Steven Scholfield, Geraldine D. Kavembe, Mahesh Nirmalan.

**Writing – original draft:** Steven Scholfield, Jared Mecha, Isaac Kibwage, Jonathan Huck, Mahesh Nirmalan.

**Writing – review & editing:** Steven Scholfield, Bernhards O. Ragama, Jared Mecha, Arpana Verma, Jonathan Huck, Diana Mitlin, Mahesh Nirmalan.

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
