## [Decision Letter · Decision Letter 0]

9 Aug 2023

PONE-D-23-14125A cross-sectional survey on the effectiveness of public health campaigns on changing knowledge, attitudes and practices during pandemics: a population-based approach in informal settlements of Kenya during the COVID-19 pandemicPLOS ONE

Dear Dr. Nirmalan,

Thank you for submitting your manuscript to PLOS ONE. After careful consideration, we feel that it has merit but does not fully meet PLOS ONE’s publication criteria as it currently stands. Therefore, we invite you to submit a revised version of the manuscript that addresses the points raised during the review process.

We look forward to receiving your revised manuscript.

Kind regards,

Megan Schmidt-Sane

Academic Editor

PLOS ONE

Journal Requirements:

"The study was funded by the Global Challenges Research Fund - University of Manchester QR Allocations"

"The study was funded by the Global Challenges Research Fund, QR allocations to the University of Manchester."

"The study was funded by the Global Challenges Research Fund - University of Manchester QR Allocations"

4. Thank you for stating the following in your Competing Interests section: "No competing interests"

5. We note that Figure 1 & Supporting Information (Appendix 1_Survey) in your submission contain map images which may be copyrighted. All PLOS content is published under the Creative Commons Attribution License (CC BY 4.0), which means that the manuscript, images, and Supporting Information files will be freely available online, and any third party is permitted to access, download, copy, distribute, and use these materials in any way, even commercially, with proper attribution. For these reasons, we cannot publish previously copyrighted maps or satellite images created using proprietary data, such as Google software (Google Maps, Street View, and Earth). For more information, see our copyright guidelines: http://journals.plos.org/plosone/s/licenses-and-copyright.

We require you to either (1) present written permission from the copyright holder to publish these figures specifically under the CC BY 4.0 license, or (2) remove the figures from your submission

(1) You may seek permission from the original copyright holder of Figure 1 & Supporting Information (Appendix 1_Survey) to publish the content specifically under the CC BY 4.0 license.  

6. We note that Supporting Information (Appendix 2 Campaign) in your submission contain copyrighted images. All PLOS content is published under the Creative Commons Attribution License (CC BY 4.0), which means that the manuscript, images, and Supporting Information files will be freely available online, and any third party is permitted to access, download, copy, distribute, and use these materials in any way, even commercially, with proper attribution. For more information, see our copyright guidelines: http://journals.plos.org/plosone/s/licenses-and-copyright.

(1) You may seek permission from the original copyright holder of Supporting Information (Appendix 2 Campaign) to publish the content specifically under the CC BY 4.0 license. 

(2) If you are unable to obtain permission from the original copyright holder to publish these figures under the CC BY 4.0 license or if the copyright holder’s requirements are incompatible with the CC BY 4.0 license, please either i) remove the figure or ii) supply a replacement figure that complies with the CC BY 4.0 license. Please check copyright information on all replacement figures and update the figure caption with source information. 

If applicable, please specify in the figure caption text when a figure is similar but not identical to the original image and is therefore for illustrative purposes only.

**Additional Editor Comments:**

Please consider the reviewers' comments, including the reviewer who requested that the authors be careful about perpetuating negative stereotypes about urban informal settlements and review the language or descriptors to ensure they are appropriate.

You may wish to consider changing the title, but this is not a necessary change.

**Comments to the Author**

1. Is the manuscript technically sound, and do the data support the conclusions?

Reviewer #1: Partly

Reviewer #2: Yes

2. Has the statistical analysis been performed appropriately and rigorously? 

Reviewer #1: No

Reviewer #2: Yes

3. Have the authors made all data underlying the findings in their manuscript fully available?

Reviewer #1: No

Reviewer #2: Yes

4. Is the manuscript presented in an intelligible fashion and written in standard English?

Reviewer #1: No

Reviewer #2: Yes

5. Review Comments to the Author

Reviewer #1: Title: Revise, too mouthful. The short title could be more concise and still convey the main message.

Key word: Compile with the journal requirement.

Abstract: Provide specific numerical results related to the observed variables. Revise grammatical errors. Methodology is missing.

Introduction: Poor structured. Limited literature review on sub-Sharan Africa, Kenya and study sites. While the introduction mentions a study in Bangladesh, it would be beneficial to discuss existing research on KAPs in informal settlements in Africa. Address the gaps in the literature that the current study aims to fill and how it expands on previous knowledge. Consider revising some sentences to make the text clearer and more focused. Ensure that the citations follow the appropriate format according to PLOS One guidelines. When discussing informal settlements and slums, be cautious about the language used to avoid perpetuating negative stereotypes. Clearly state the specific objectives of the current study in the introduction. Consider including more recent statistics or relevant information on the prevalence of COVID-19 in these settlements.

Methodology: Is it quantitative, qualitative or mix method study. Sampling should be a different topic from study design and elaborate more on sampling procedures. While it is mentioned that the population estimates were made in-house for the study, it would be beneficial to elaborate on the methodology used to estimate the population in each settlement. Mention the sample size calculation and justify it. Explain the randomization process for survey administration in more detail. Provide a more detailed explanation on data collection procedure that ensure the confidentiality and privacy of the participants. Justification for choosing study sites not clear, while the health promotion activities are outlined, it would be helpful to elaborate on the strategies used to evaluate the effectiveness of these interventions. Did the study team collect data on the reach and impact of the health promotion campaign? What are study dependent and independents variables? Monitoring of field study should be a different topic from ethical consideration. Data management is lacking.

Results: The observation of potential differences in KAPs between settlements is intriguing, discuss potential reasons for these differences, The authors mentioned that 39 participants were excluded from data analysis due to various reasons, such as unfeasible answers, contradictory responses, and incorrect data entries. It would be helpful to provide more specific examples of such issues to understand their impact on the data and the potential implications for the study's conclusions. In socio-demography why separate college, university and post-degree instead of summing it up as tertiary education. Which variable are compared against to generate the P-values indicated.

Discussion: Should be based on the study's findings. Expound on the discussion of the findings and addressing any limitations will strengthen the research's impact and implications for public health efforts. Consider providing more interpretation and discussion of the observed changes in behaviors, attitudes, and perceptions. Are there any factors that might have contributed to these changes? How do the findings align with previous research or public health campaigns? Discuss the practical implications of these changes for public health interventions.

Conclusion: Should be based on the study's findings. What are the key takeaways for policymakers and public health practitioners in Kenya or similar regions facing similar challenges?

Language and Clarity: The manuscript's language should be revised for clarity and precision. Some sentences appear convoluted and may benefit from rephrasing. Additionally, ensure consistent use of terminology throughout the manuscript.

Reviewer #2: Very good paper.

Abstract should briefly mention the study sites for ease of understanding for the reader.

Statistical analysis: please clarify with analysis package was used, is it SPSS or STATA? the paper mentions both as(SPPS or STATA)

The intervention was community education through various means including radio, use of CHVs and health providers, however, given the abundance of information at that time, the authors should mention the possible role/influence/limitations of social media on the outcomes.

Methods: Its mentioned, the population was stratified, and purposively selected, from outdoor participants, how did the study team ensure that the was no bias with only those interested in the study taking part? how were the two participants identified to reduce bias?

The age range should also be mentioned for a better understanding of the study population?

Details on Mask distribution: Given the population in the informal settlements, how were the 60,000 masks distributed ? how does that relate to the 720 respondents access to the masks? were the 720 respondents targeted to receive the masks? being repeated cross sectional surveys, the respondents at the at baseline and endline were different, how does that relate to the access to the distributed masks?

Was Consenting part of the process? please indicate under ethics.

6. PLOS authors have the option to publish the peer review history of their article (what does this mean?). If published, this will include your full peer review and any attached files.

Reviewer #1: **Yes: **Ismail Adow Ahmed

Reviewer #2: No

---

## [Author Response · Author response to Decision Letter 0]

21 Sep 2023

All observations made by referees 1 & 2 have been fully addressed. These changes are shown in the version of the manuscript where the changes are Tracked.

---

## [Editor Report · Decision Letter 1]

27 Oct 2023

A cross-sectional survey on the effectiveness of public health campaigns for changing knowledge, attitudes, and practices in Kenyan informal settlements during the COVID-19 pandemic

PONE-D-23-14125R1

Dear Dr. Nirmalan,

We’re pleased to inform you that your manuscript has been judged scientifically suitable for publication and will be formally accepted for publication once it meets all outstanding technical requirements.

Kind regards,

Megan Schmidt-Sane

Academic Editor

PLOS ONE

---

## [Editor Report · Acceptance letter]

8 Nov 2023

PONE-D-23-14125R1 

A cross-sectional survey on the effectiveness of public health campaigns for changing knowledge, attitudes, and practices in Kenyan informal settlements during the COVID-19 pandemic

Dear Dr. Nirmalan:

I'm pleased to inform you that your manuscript has been deemed suitable for publication in PLOS ONE. Congratulations! Your manuscript is now with our production department. 

Kind regards, 

on behalf of

Dr. Megan Schmidt-Sane 

Academic Editor

PLOS ONE